# Dual-color emissive OLED with orthogonal polarization modes

Ruixiang Chen[1], Ningning Liang ◉[1] ✉ & Tianrui Zhai ◉[1] ✉

Linearly polarized organic light-emitting diodes have become appealing functional expansions of polarization optics and optoelectronic applications. However, the current linearly polarized diodes exhibit low polarization performance, cost-prohibitive process, and monochromatic modulation limit. Herein, we develop a switchable dual-color orthogonal linear polarization mode in organic light-emitting diode, based on a dielectric/metal nanograting-waveguide hybrid-microcavity using cost-efficient laser interference lithography and vacuum thermal evaporation. This acquired diode presents a transverse-electric/transverse-magnetic polarization extinction ratio of 15.8 dB with a divergence angle of ±30°, an external quantum efficiency of 2.25%, and orthogonal polarized colors from green to sky-blue. This rasterization of dielectric/metal-cathode further satisfies momentum matching between waveguide and air mode, diffracting both the targeted sky-blue transverse-electric mode and the off-confined green transverse-magnetic mode. Therefore, a polarization-encrypted colorful optical image is proposed, representing a significant step toward the low-cost high-performance linearly polarized light-emitting diodes and electrically-inspired polarization encryption for color images.

Organic light-emitting diodes (OLEDs) have many distinct merits, such as a high contrast ratio, wide viewing angle, self-luminescence, low driving voltage, flexibility, lightweight, and so on, which have promoted their commercialization for full-color display panels, eco-friendly lighting sources as well as the most promising technology in solid-state light source, optical data communications, and photonics[1–10]. As an important freedom of light waves, the linearly polarized characteristic endows OLEDs with controllable spatial pattern of the emission spectra, which is completely distinguished from the conventional OLED based on ITO that a Lambertian-like air mode emission with a non-directional emission[11,12]. This characteristic affords linearly polarized organic light-emitting diodes (LP-OLED) with great potentials for full resolution autostereoscopic naked-eye three-dimensional displays, visible light communication, and information anticounterfeiting and encryption, inspiring various efforts been explored for the LP-OLEDs[13–17].

The major obstacle for LP-OLED is the acquisition of a high linear polarization extinction ratio (ER)[18]. As for LP-OLED, organic single-crystal-based OLED presents an inherent in-plane anisotropic nature for direct LP light emission. Nevertheless, compared to the thin-film LP-OLED, these organic single-crystal-based ones are becoming incompetent for the construction of large-area LP electroluminescence devices[19,20]. In these planar OLEDs, some photons are confined within the substrate and waveguide layer, while propagating along the organic/metal interface[21]. As a result, the leak into air acts as background light, leading to complex emission profiles with poor polarization extinction ratio and low light extraction efficiency (LEE)[22]. This ratio is defined as $ER_{TM/TE} = 10 log \frac{I_{TM}}{I_{TE}}$, where $I_{TM}$ and $I_{TE}$ are intensities of transverse-magnetic (TM) and transverse-electric (TE) polarized emission at the same wavelength. The first LP-OLED can be tracked into 1995, in which, Dyreklev et al. introduced aligned conjugated polymers as the emitting layer, endowing the resulting device

[1]School of Physics and Optoelectronic Engineering, Beijing University of Technology, Beijing 100124, China. ✉e-mail: liangnn2020@bjut.edu.cn; trzhai@bjut.edu.cn

with a polarization anisotropy of 3.1, but with an EQE as low as 0.1%[23]. Consequently, several orientation methods, like friction-transfer processing, tensile alignment, Langmuir-Blodgett deposition, and pre-aligned substrates[24–26], have been developed rapidly for the emission of linearly polarized light; however, these methods usually damage the active layer and result in a wide emission peak with low color purity, as well as a low ER. An alternative way to directly capitalize a planar OLED on an integrated nanograting could be to precisely regulate the internal light field distribution and achieve a high ER[27–33]. Despite this, these externally integrated optical structures, such as birefringent photonic polarizers, generally result in at least 50% light loss and a large emission angle[34,35], and result in passively modulating optical properties for a narrowed bandwidth and a resulting pure linear polarized color, owing to weak light-matter interactions[36,37].

Therefore, an effective strategy should be adopted to improve the comprehensive performance of LP-OLED, and the method of embedding a nanograting structure into a stacked OLED device has recently been identified as a straightforward and efficient approach[38,39]. Moreover, for a planar device, the TM waveguide mode apparently excites the surface plasmon polariton (SPP) mode that distributes near the metal surfaces and decays quickly toward the middle of the device, causing a large emission loss. By contrast, the TE waveguide mode polarized parallel to the thin-film surface with different directional propagations can be extracted significantly by constructing a corrugated structure. In 2021, So et al. demonstrated a LP-OLED with a small divergence angle less than 3°, a moderate $ER_{TE/TM}$ of 11.1 dB at ~520 nm and a high EQE of 7%, providing one of the few successful cases of polarization modulation by embedding an optical cavities using a combination of interference lithography and reactive ion etching[39]. Notably, improving the EQE, increasing the ER, realizing a small divergence angle, and developing a cost-efficient method to replace the high-cost and time-consuming fabrication process, which involves electron beam etching, ion etching, and nano-imprinting lithography,

are the major challenges associated with the developing of internally integrated LP-OLED devices[40,41]. All the above-mentioned issues inevitably hinder the commercialization process, especially for large-area LP-OLED. Furthermore, considering dynamic holographic displays and polarized colorful image encryption, multiwavelength modulation with different polarization modes in a single OLED device is in high demand, but has not been reported[42–44].

In this study, one-dimensional dielectric/metal (D/M) nanogratings as large as 3 × 3 cm² embedded in OLEDs are obtained through a combined low-cost methods of laser interference lithography (VIL) and vacuum thermal evaporation (VTE). By optimizing the electron transport layer (ETL) thickness, the TE waveguide mode at the desired wavelength is significantly confined within the metal reflection waveguide (MRW). Consequently, the high-intensity TE waveguide mode is diffracted into the air free space, owing to the formation of the corrugated OLED. Finally, an OLED with dual-color orthogonal polarized emission modes is achieved. It exhibits a TE/TM polarization ER of 15.8 dB, a suppressed full width at half maximum (FWHM) of 28 nm, a divergence angle of ±30° for sky-blue light, along with a green TM mode light emission. These devices provide an excellent theoretical strategy for the polarization-encryption of colorful images. The proposed design concept can be extended to full-color gamut linearly polarized modulation with high ER and excellent EQE, providing a powerful platform for manufacturing low-cost, large-area, multi-color emissive LP-OLED with multiple polarizations for three-dimensional displays, augmented/virtual reality, high-density data storage, and optical encryption.

## Results

### Design and construction of dielectric/metal nanograting
As shown in Fig. 1a, to form a highly polarized emission with a narrow emission cone, corrugated OLED structures embedded with a dielectric/metal (D/M) nanograting geometry were designed and constructed, through finite-difference time-domain (FDTD) simulation and combined

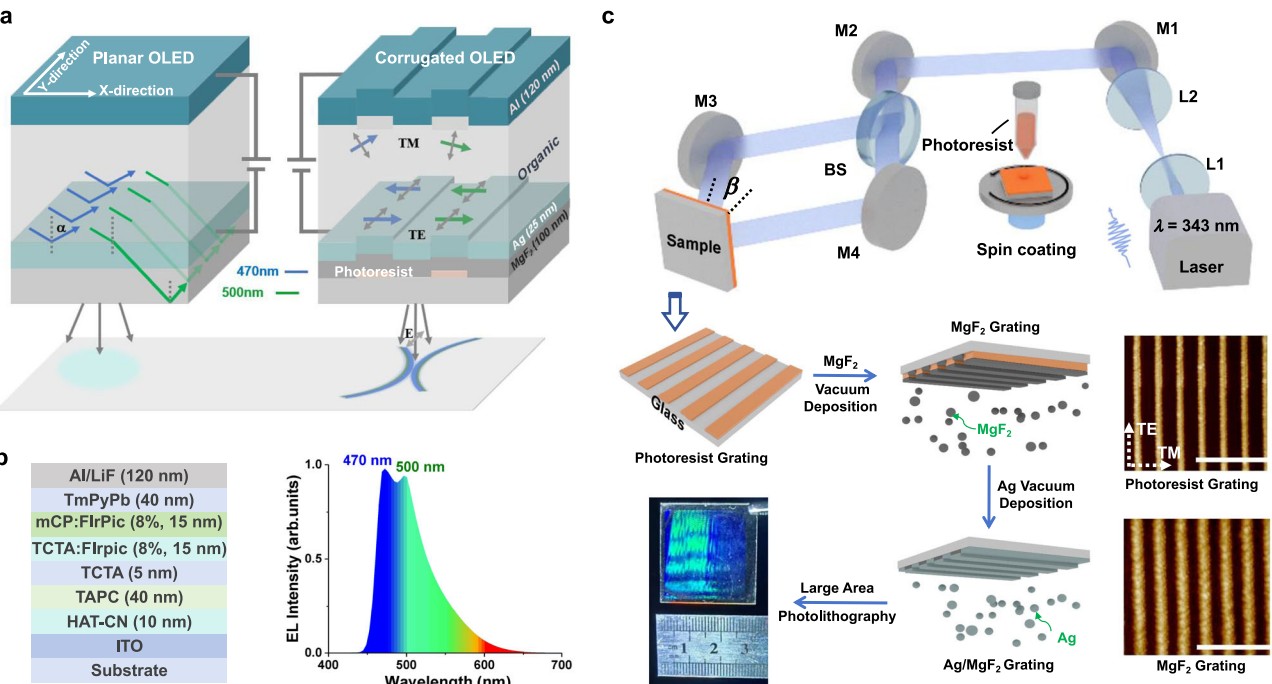

**Fig. 1 | Design idea and illustration of the fabrication process for dielectric/metal nanograting. a** Schematic of planar and corrugated OLED with different emission patterns. Blue corresponds to a wavelength of 470 nm, while green corresponds to a wavelength of 500 nm. **b** Device architecture and electroluminescence (EL) spectrum of conventional OLED with the same FIrPic-based emitting layer (EML), adopting ITO as anode[45]. **c** Experimental setup and fabrication process of dielectric/metal nanograting. Transverse-electric (TE) polarization represents the polarization direction of light that is parallel to the direction of the grating, while transverse-magnetic (TM) polarization represents the polarization direction that is perpendicular to it. Scale bar, 1 μm. Here L, M, and BS correspond to convex lenses, reflectors, and beam splitters, respectively.

methods of laser interference lithography and vacuum thermal evaporation. In this study, the structure of a glass substrate/cathode/electron injection layer (EIL)/electron transport layer (ETL)/emitting layer (EML)/hole transport layer (HTL)/hole injection layer (HIL)/anode (120-nm-Al) was utilized to construct an inverted bottom-emitting OLED. The cathode for corrugated OLED was composed of a D/M nanograting with a structure of photoresist/100-nm-MgF$_2$/25-nm-Ag; and a planar 25-nm-Ag film was fabricated for the planar OLED as a control group. The EML layer was deliberately composed by the commercialized host materials, N,N'-dicarbazolyl-3,5-benzene (mCP) and 4,4'4"-tris(N-carbazolyl) triphenyl-amine (TCTA) and blue emissive guest materials Ir-complex iridium (III)bis-(4,6-difluorophenyl-pyridinato-N,C$^2$)-picolinate (FIrPiC) with dual layers of TCTA:FIrPic (8% wt) and mCP:FIrPiC (8% wt)[45]. The molecular structures of emitting materials are shown in Supplementary Fig. 1 and corresponding EL spectrum for conventional OLEDs employing the same FIrpic-based EML adopting indium tin oxide (ITO) as anode ranges from 440 to 650 nm, which exhibits a FWHM of 67 nm and a dual-peak at 470 and 500 nm (Fig. 1b), respectively[45].

To produce a regular pattern, a maskless laser lithographic technique[46] was employed, using an interference pattern from two oblique incident beams at a wavelength of 343 nm to illuminate the spin-coated photoresist film, as shown in Fig. 1c. With the illumination and development of the photoresist, the interference pattern of coherent light was recorded in the photoresist. After the subsequent vacuum thermal evaporation of MgF$_2$ with a thickness of 100 nm and a thinner 25-nm-Ag film, a D/M nanograting was successfully transferred to the cathode. The period of the interference pattern and thus that of the grating recorded on the substrate is given by the grating diffraction equation (Supplementary Note 1). Herein, the thickness of the bottom Ag layer was reasonably set to 25 nm with a refractive index ($n$) of 0.13 and an extinction coefficient ($k$) of 2.66 (Supplementary Fig. 2a) to ensure the conductivity and transparency of the transparent electrodes for OLEDs. To optimize the dielectric material, the simulated far-field distribution of the corrugated OLED in the normal direction was calculated (Supplementary Fig. 2b) and the resulting output intensity reaches its maximum when the effective refractive index ($n_{eff}$) is 1.60. Considering the $n$ for photoresist is -1.70, along with the equivalent refractive index theory, the MgF$_2$ with a lower $n$ (1.38 at $\lambda = 470$ nm) and lower optical loss for blue light region, was deliberately incorporated. Consequently, a 100-nm-thick MgF$_2$ layer was deposited on the corrugated photoresist film to ensure a sufficiently deep groove. However, the relatively low $n$ of the substrate is reasonable for suppressing the substrate mode and enhancing the light-extraction efficiency of linearly polarized light.

Via precisely controlling the recording angle $\beta$, laser power, exposure time and developing time of photoresist film, along with the thickness of MgF$_2$ layer, a MgF$_2$/photoresist grating with a certain groove depth of 80 nm, a period of 300 nm, and a ridge width of 150 nm, was successfully realized for the intracavity resonance of 470 nm wavelength, As presented in Fig. 1c. Excitingly, via modulating the $\beta$ value, a large-area Ag/MgF$_2$/photoresist grating was realized with a size of 3 × 3 cm$^2$, when $\beta$ was set at 34.9°. This combined LIL and VTE method is an excellent, relatively easy, and cost-efficient technique for fabricating large-area corrugated OLED.

## Confinement of transverse-electric waveguide in planar OLEDs

Considering the rare reports on sky-blue LP-OLED with high ER and EQE, one of the main tasks of this work is focused on the efficient coupling output of 470 nm light for high polarization with a narrow linewidth. To achieve pure TE waveguide emission at a wavelength of 470 nm, the desired geometric structure of the stacked OLED should be rationally designed to completely suppress the emission of the TM waveguide, SPP, substrate, and air modes, along with the efficient diffraction of the localized TE waveguide mode using a corrugated structure. From the measured reflection and transmission spectra shown in Supplementary

Fig. 3a, reflectivity as high as 70% and 97% are obtained from the bottom 25-nm-Ag cathode and the 120-nm-Al anode, respectively. In addition to the sandwiched organic layer, a planar Fabry-Pérot (F-P) nanocavity was successfully constructed, thereby forming a MRW and improving mode confinement within the gain-active medium[47].

To achieve the maximum confinement of the waveguide mode, reasonably enlarging the effect refractive index of waveguide layer was necessary. Because of the relatively higher $n$ of ETL (TmPyPb) than that of the HTL (TAPC), the ETL thickness was deliberately regulated. First, the intensity and corresponding proportional changes of different modes at $\lambda_0$ of 470 and 500 nm, including the air mode, substrate (Sub) mode, absorption mode, surface plasmon (SPP) mode, and waveguide (WG) mode, as the ETL thickness increased, were simulated for planar OLED (Fig. 2a, b). Notably, the air-mode emission at 470-nm-wavelength is suppressed almost completely when the ETL thickness ranged from 30 nm to 115 nm, combined with an increasing SPP mode and decreasing substrate and waveguide modes. Similarly, the air mode at $\lambda_0 = 500$ nm presents an analogous tendency, but with a slightly suppressed valley and right-shift, which is mainly ascribed to the difference in wavelength response to the F-P cavity[48]. To clearly understand the influence of the MRW geometry on the mode distribution for TE and TM polarization at a wavelength of 470 nm, the far-field angular emission patterns of both polarization modes for the planar OLED with different ETL thicknesses were simulated, and the corresponding emission intensities are represented by a left dashed line in the illustration of Fig. 2c, d. The TE mode at various ETL thicknesses is emitted at a specific angle with complete confinement in the waveguide in the ETL thickness range of 34 –113 nm. The observed emission pattern and change in intensity can be illustrated using the following formula[49,50]:

$$\frac{4\pi n_{eff} h cos\alpha}{\lambda_0} - \Psi_1 - \Psi_2 = m_1 2\pi \qquad (1)$$

where $\alpha$ is the incident angle of the photons on the metal film, $h$ the cavity length or organic WG thickness, $n_{eff}$ is the effective refractive index, $m_1$ the order and $\psi$ reflection phase shift. A detailed analysis is provided in Supplementary Note 2. Here, $n_{eff}$ is determined as 1.51 when the 107-nm-thick ETL with $n = 1.81$ and EML with $n = 1.75$ (Supplementary Fig. 3b). As shown in Supplementary Fig. 4, for any frequency photons that propagate at a certain angle $\alpha$ between two electrodes, only when the optical path difference between two adjacent outgoing beams satisfies the coherent phase length condition does, the modes finish the emitting behavior with a uniquely determined view angle $\alpha_1$ that is equal to $arcsin\frac{n_{eff}}{sin\alpha}$. For TE mode at $\lambda = 470$ nm, its view angle is enlarged as the increased ETL thickness, presenting a nearly horizontal emission by breaking phase matching at the case of a matched thickness of 107 nm (Fig. 2h, see below). In this case, the TM emission is presenting a profile. This is ascribed to the fact that the TE mode was completely confined to the waveguide layer (Fig. 2e), whereas a large section of the TM waveguide mode excites SPP mode, resulting in associated electromagnetic fields located at the Ag/organic interface (Fig. 2f).

To separately determine the optimal ETL thickness for the maximum confinement of TE polarized mode into waveguide layer, all mode intensities at $\lambda = 470$ and 500 nm as the function of ETL thickness were simulated. As shown in Fig. 2g, h, multimode photon localization is successfully controlled by modulating the waveguide (ETL) thickness; for instance, when the ETL is controlled at 107 nm, these unwanted extra modes (Sub and air mode) are suppressed to the maximum extent, thereby confining the target single TE mode at the desired 470 nm wavelength.

## Transverse-electric mode extraction in corrugated OLEDs

Based on the above simulation and analysis, a reasonable Bragg grating geometry was embedded to achieve TE polarized light extraction. Therefore, another task in this work is to fine-tune the geometric

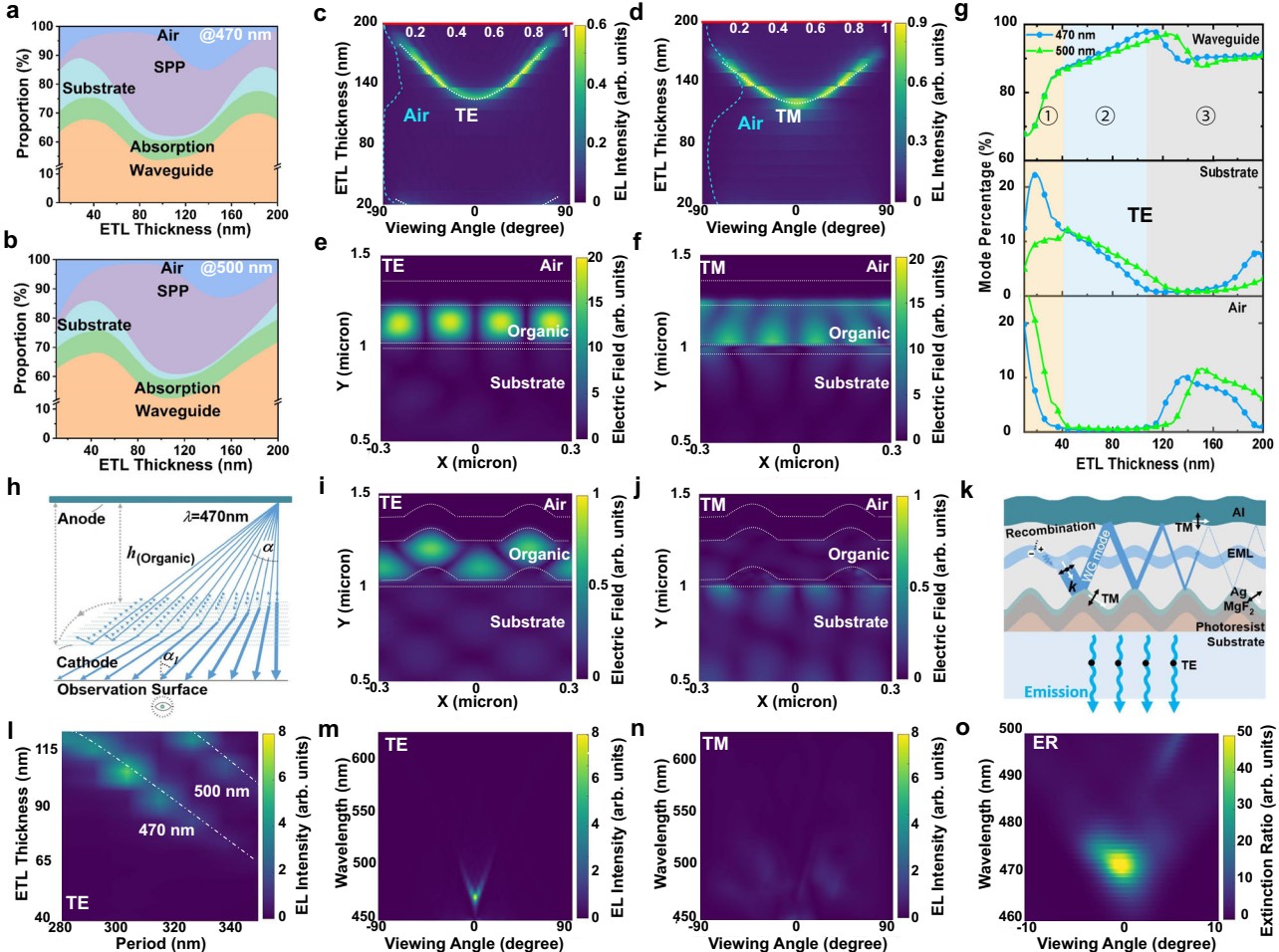

**Fig. 2 | Simulation diagram of optical properties of planar and corrugated OLED.** Simulated optical mode distribution of planar OLED at the wavelength of **a** 470 nm and **b** 500 nm as the electron transport layer (ETL) thickness, including air, surface plasmon polariton mode (SPP), substrate (Sub), waveguide (WG), and absorption. The corresponding angular electroluminescence (EL) emission of **c** transverse-electric (TE) and **d** transverse-magnetic (TM) polarization at 470 nm. The insert shows the corresponding air mode distribution, with its intensity marked at the top of the figure. The electromagnetic field distribution of **e** TE and **f** TM polarization modes at 470 nm for planar OLED. **g** Simulated TE polarization optical mode distribution of planar OLED at 470 (blue) and 500 nm (green) wavelength as the ETL thickness. **h** Schematic of planar OLED light emission with a gradual increase in emission angle as a more waveguide thickness, for a certain wavelength of 470 nm. The electromagnetic field distribution of **i** TE and **j** TM polarization modes at 470 nm for corrugated OLED. **k** Schematic of the corrugated OLED. **l** Far-field electric field intensity of corrugated OLED as grating period and ETL thickness, when the emission angle is 0°. Simulated far-field angular emission of TE (**m**) and TM (**n**) polarization modes for corrugated OLED, and (**o**) the corresponding simulated extinction ratio (ER) for $10log\frac{I_{TE}}{I_{TM}}$.

parameters of the D/M nanograting-waveguide structure to realize a switchable dual-color OLED with highly orthogonal polarization modes. A corrugated OLED structure was constructed with an ETL thickness of 107 nm, groove depth of 80 nm, period of 300 nm, and ridge width of 150 nm for the photoresist/MgF$_2$ nanograting. The shape of the grating unit was adopted according to the AFM and cross-section SEM results for the D/M grating and OLED device, respectively.

The resulting simulated electric-field intensity contours for plane wave incidence in the TE and TM modes for the corrugated LP-OLEDs are shown in Fig. 2l, j. The TE-mode light is transmitted into the air free space, whereas TM-polarized mode is transformed by the grating, forming a certain SPP mode, as schematic in Fig. 2k. This phenomenon is consistent with the simulated position distributions of the TE and TM modes (Supplementary Fig. 5a), most of the TM modes are located at the interface of the metal and dielectric, generating the SPP mode, and the TE mode forms the waveguide mode. Moreover, because of the strong quenching by the metal layer, the TM mode experiences severe attenuation[51] and only the TE mode intensity changes periodically with the ETL thickness. Herein, we define the light extraction

efficiency ratio (LEER) between planar and corrugated OLED, as LEER = $\frac{LEE_{corrugated}}{LEE_{planar}}$. For TE-polarized mode, LEER approaches ~ 15 times when the ETL thickness is determined to be 107 nm, whereas it is close to zero for TM-polarized mode (Supplementary Fig. 5b). Essentially, this phenomenon results from the momentum match between the TE waveguide light and the air photon. The Bragg gratings efficiently promote out-coupling by providing the TE waveguide light with additional momentum to couple into the air space. The momentum matching condition is given by the Bragg equation[52]:

$$\mathbf{k}_0 = \mathbf{k}_{wg} \pm m_2\mathbf{k}_G \qquad (2)$$

here $\mathbf{k}_0 = 2\pi/\lambda_0$, $\mathbf{k}_{wg} = 2\pi n_{eff}/\lambda_0$, $\mathbf{k}_G = 2\pi/\Lambda$ is the momentum of light in air, waveguide, and grating, respectively; $m_2$ is the diffraction order. For a given wavelength $\lambda_0$, the TE waveguide mode coupled to air could be tuned by varying the grating period, effective refractive index, and waveguide layer thickness (Fig. 2l). Herein, the intensity and wavelength of emitted TE waveguide mode into air are governed by the ETL

thickness and grating period Λ, that overcomes the phase matching of F-P cavity and meanwhile satisfies the momentum matching of grating structure. Therefore, a 300-nm-period grating is adopted for the 107-nm-thick ETL to extract the strongest waveguide mode at 470 nm. This negligible TM diffraction mode is ascribed to the polarization selectivity of the grating structure for the waveguide mode, along with weak coupling between the momentum of $\mathbf{k_{SPP}}$ and $\mathbf{k_G}$.

Furthermore, the mode dispersion for TE and TM polarization mode presents a distinct diffraction profile with a maximum intensity and negligible emission intensity at $\lambda = 470$ nm, respectively (Fig. 2m, n). The simulated $ER_{TE/TM}$ reveals the excellent linear polarization characteristic of the optimized LP-OLED with a high ER, a small divergence angle and a narrow bandwidth at $\lambda = 470$ nm (Fig. 2o). This is ascribed to the fact that the corresponding waveguide mode at the desired wavelength breaks the phase matching of the F-P cavity and therefore matches the momentum condition of the grating geometry in the case of the optimized ETL thickness and grating period.

### LP-OLEDs property characterization and analysis

To validate our design concept, we fabricated corrugated OLEDs with a grating period of 300 nm and ETL thickness of 107 nm to investigate the effect of the D/M nanograting-waveguide geometry on device performance, including the EQE, ER, and electroluminescence angular dispersion characteristics. In this study, a corrugated OLED with an inverted structure of glass/photoresist-MgF$_2$-Ag nanograting/LiF/TmPyPb/mCP:FIrPiC/TCTA:FIrPiC/TCTA/TAPC/HATCN/Al (Fig. 3a) was fabricated. According to Fig. 2g, it clearly demonstrated that both a 60-nm-thick ETL and a 107-nm-thick ETL effectively suppress TE air-mode emission, with a higher localization of waveguide modes for OLED based on 107-nm-thick ETL. Therefore, planar inverted OLEDs with an ETL thickness of 107 nm and 60 nm were also fabricated for comparison and were named as planar and reference OLED, respectively. According to the cross-sectional scanning electron microscope (SEM) images of these multilayer (Fig. 3b), a significantly corrugated geometry is clearly observed until the top thick Al anode for the stacked OLED. Elemental map reveals its notable structural characteristics.

The EL spectra, luminance-voltage-EQE curves, angle-resolved EL intensity in the normal direction at an emission peak of 470 nm wavelength, corresponding TE-and TM-polarized mode emission, and $J$-$V$ curves were measured using the same setup, as shown in Fig. 3c–l and Supplementary Fig. 6. Due to the distinct directional emission of the corrugated OLED, breaking the traditional Lambertian emission mode, an integrating sphere was deliberately utilized to characterize the device $J$-$V$ curves, EL spectra, and luminance-voltage-EQE curves, aiming to collect all the photons emitted from the front and back, as well as from sides of the device[53,54]. Interestingly, the corrugated OLED exhibits a significantly enhanced current density than that of planar OLED with 107-nm-ETL. Contrast to the reference OLED with 60-nm-ETL, both planar and corrugated OLED with 107-nm-ETL, presents a significantly narrowed emission spectra with a distinguishable EL peak at $\lambda$ ~ 470 nm and a significantly suppressed peak at $\lambda$ ~ 500 nm (Fig. 3c). These results indicate that stronger TE waveguide localization, rather than air mode suppression, is more beneficial for the emission of TE polarized light. The optimized corrugated OLED displays a suppressed FWHM of 28 nm, from 205 nm for the reference OLED, with a CIE from (0.31,0.48) to (0.13,0.35) (Fig. 3d). This clearly demonstrates that the successful introduction of the D/M nanograting waveguide geometry significantly shifted the EL spectra to desired blue-sky region. From Fig. 3e, the turn-on voltages ($V_{on}$) of the optimized LP-OLED are also suppressed to 4.7 V and the corresponding luminance exceeds 48 cd m$^{-2}$ at 8 V, significantly higher than that of 7 cd m$^{-2}$ for planar OLED. The EQE values of the planar and corrugated OLED devices are measured to be 1.23% and 2.25% at 8 V, respectively. The detailed parameters of all devices are summarized in Table 1. Notably, the optimized corrugated OLEDs exhibits a lower EQE than

conventional planar ITO-based OLED with the same FIrpic-based EML based on ITO in (Supplementary Fig. 7a, b), which results from the employed 25-nm-Ag cathode and ultrathick 107-nm-ETL; whereas, a TE polarization mode emission is realized (Fig. 3f). Detailed characterization and analyses are presented in the following sections. Furthermore, the corrugated LP-OLED exhibits relatively higher storage stability than the planar and conventional OLED, as summarized in Supplementary Fig. 8. This may be mainly ascribed to the hybrid microcavity effect on the exciton lifetime and a more uniform photons distribution[55,56].

We experimentally assessed the performances of the emitted TE-and TM-polarized mode dispersions by employing angle-resolved emission spectra measurements by rotating the polarizer $\theta$ (Supplementary Fig. 9 and Supplementary Methods). For the planar OLED, the air mode is significantly suppressed, with slight TE mode emission at wavelengths ranging from 460 to 500 nm (Fig. 3g–i). The EL intensities at both specific wavelengths are deliberately extracted revealing weakly polarized emissions at both wavelengths. Excitingly, the optimized corrugated OLED presents a high $ER_{TE/TM}$ of 15.8 dB at $\lambda = 470$ nm, small divergence angle of ±30°, and a narrowed FWHM of 28 nm (Fig. 3f, j and k). This is consistent with the simulation results as Fig. 2m. In contrast, there are almost no waveguide or SPP mode diffraction features in the TM mode at the emission peak of 470 nm (Fig. 3l). Notably, the polarization ratio of the linearly polarized light source required for commercialization is 14.7–16.0 dB, which indicates the commercial application potential of our devices.

### Dual-color emission with orthogonal polarization modes

Accordingly, for the optimized LP-OLEDs, the TE polarized mode presents a sky-blue color, with a $\theta$ of 90°, as shown in Fig. 4a and Supplementary Movie 1; whereas, the intensity of sky-blue TE polarized light drops dramatically as the polarizer was rotated to 0°, ultimately displaying a green TM polarized light (Fig. 4b). This creates a dual-color emissive OLED with an orthogonal polarization mode. Furthermore, the resulting emission pattern exhibits a dual-intersecting bright arc corresponding to the two back-propagation waveguide modes of grating diffraction, as displayed in Fig. 4c, which is completely different from the Lambertian profile of the planar OLED, as shown in the inserted map. The emission profile of the corrugated OLED displays both horizontal and vertical symmetry, resulting from the symmetry of grating structure and exciton recombination randomness[57,58]. Moreover, the inner side of the arc displays a high sky-blue luminance, whereas a low green luminance is observed on the outer side. This can be explained using reciprocal space. Compared with the simulated mode dispersion of dielectric materials for planar OLED devices (Fig. 4d), the total internal reflection and dispersion formed by the waveguide (WG), substrate (Sub), and air modes are significantly modulated by the insertion of the grating momentum (Fig. 4e). Owing to the additional momentum $\mathbf{k_G}$ by the nanograting to couple the light within the waveguide into the air, two dual-colorful arcs are observed with a short wavelength in the inner side for air dispersion, with a stronger resonance and higher emission intensity when approaching the arc center. This is consistent with the experimental results presented in Fig. 4e.

To further assess this statement quantitatively, a detailed comparison with state-of-the-art LP-OLED is summarized in Supplementary Table 1, where the negative data represented $10log\frac{I_{TM}}{I_{TE}}$. Figure 4f depicts a comparison of the LP-OLED emitting the TE polarization mode via the strategies of molecular alignment and external and embedded optical nanostructures. It is clearly demonstrated that all these parameters are among the best for LP-OLED devices. In addition, this attempt to employ an organic semiconductor for a dual-color emissive OLED with orthogonal linearly polarized modes could be further utilized to design LP-OLEDs with a whole visible color gamut for high TE polarization (Fig. 4g).As the ETL thickness increases from 100 to 200 nm, combined with the corresponding period at 300–400 nm,

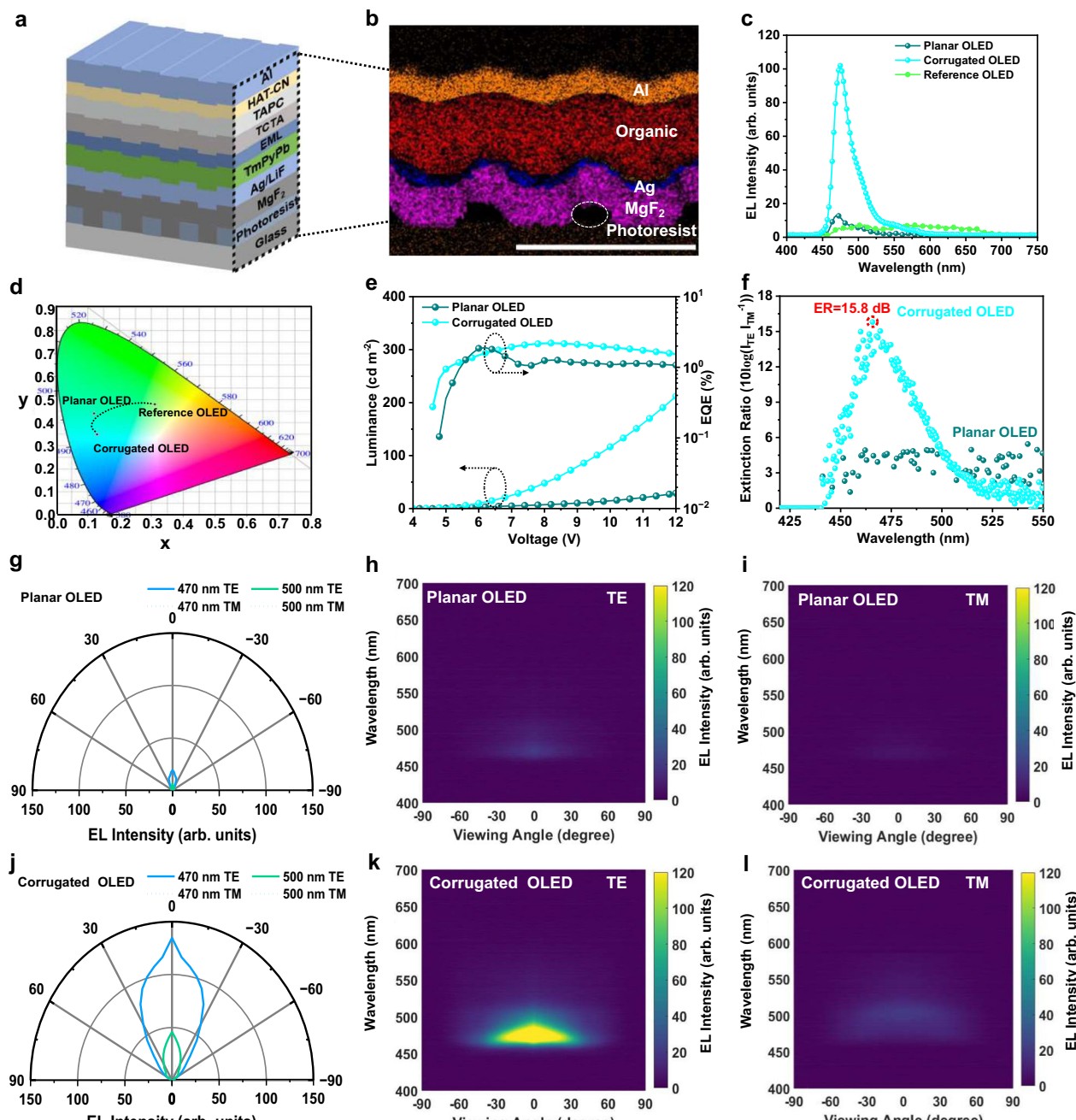

**Fig. 3 | Experimental results for planar and corrugated OLED. a** Device geometry of corrugated OLED. **b** Cross-section scanning electron microscope (SEM) of the corrugated OLEDs with 107 nm electron transport layer (ETL). Scale bar, 500 nm. **c** Electroluminescence (EL) spectra, **d** standard CIE 1931 chromatic coordinates diagram, **e** Luminance- external quantum efficiency (EQE)-voltage curves, and **f** polarization extinction ratio for these planar and corrugated OLEDs. Cyan corresponds to corrugated OLED, dark blue corresponds to planar OLED, green corresponds to reference OLED, and the extinction ratio (ER) is marked in red. **g–i** Measured mode dispersion and mode dispersion in transverse-electric (TE) and transverse-magnetic (TM) polarizations for planar OLEDs and **j–l** corrugated OLEDs with 107 nm ETL thickness. Blue corresponds to a wavelength of 470 nm, while green corresponds to a wavelength of 500 nm. The TE polarization is represented by a solid line, while the TM polarization is represented by a dashed line.

**Table 1 | Summary performance parameters of planar and corrugated OLED devices with electron transport layer thickness of 107 nm**

| Device | $V_{on}$ (V) | EL intensity (arb. units) | FWHM (nm) | $\eta_{C,max}$ (cd A$^{-1}$) | $\eta_{p,max}$ (lm W$^{-1}$) | ER$_{max}$ (dB) | EQE MAX/at 8 V (%) |
|---|---|---|---|---|---|---|---|
| Planar OLED | 5.2 | 13 | 38 | 4.15 | 2.33 | 4.9 (@500 nm) | 1.93/1.23 |
| Corrugated OLED | 4.7 | 103 | 28 | 3.81 | 1.60 | 15.8 (@470 nm) | 2.25/2.23 |

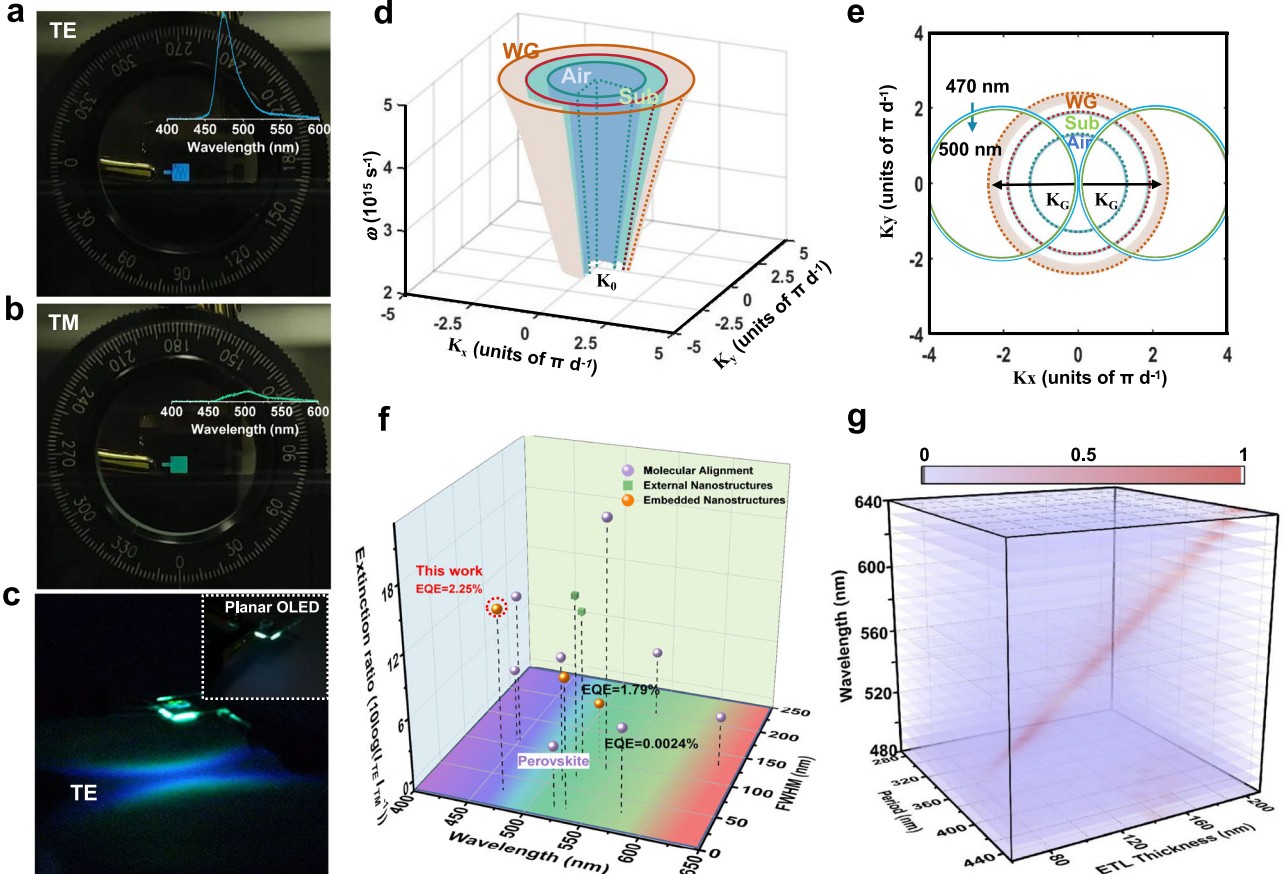

**Fig. 4 | Spatial pattern displays of single corrugated OLED and developing status. a** Transverse-electric (TE), **b** transverse-magnetic (TM) polarization electroluminescence and **c** corrugated OLED emission spatial patterns. All devices were driven at 3 mA cm⁻². **d** Schematic drawing of the simplified 3D optical modes consisting of the air mode (Air, blue), waveguide mode (WG, red) and substrate mode (Sub, green) **e** These three modes were diffracted by **k**$_G$ and −**k**$_G$ to form the two diffracted WG modes. **f** Summary performance parameters of LP-LED for TE polarization mode emission. Purple ball represent molecular alignment, green square represents external nanostructures, and yellow square represent embedded nanostructures. **g** Simulated coupling conditions at any wavelength for TE polarization mode emission.

the desired colorful emission ranging from 480 to 640 nm is achieved. Therefore, another narrow-bandgap OLED with red TE-mode and green TM-mode light could also be achieved, in which the grating period and ETL thickness are controlled at 385 nm and 192 nm, respectively. Accordingly, the dynamic modulation of dual-color polarized light emission at any wavelength with a high EQE, narrow bandwidth, strong polarization, and small divergence angle could be effectively realized by simply matching the emission wavelength, grating period, and ETL thickness.

### Electroluminescent linear polarization application

Now that we have established these high-performance LP-OLEDs with dual-color orthogonal polarization, the distinct emission behavior inspires us to further explore their applications in color-image encryption. The connection between electroluminescence and keys in color image encryption applications is presented in Fig. 5. As shown in Fig. 5a, a full-color display can be achieved by combining the two types of corrugated OLEDs. As the optical axis of the polarizer is rotated from 0° to 90°, two dual-color orthogonal polarization transitions are observed, one from green TM-polarized light to blue TE light for the wide-bandgap OLED and the other from green TM-polarized light to red TE light in the narrow-bandgap device. Taking *Van Gogh's Iris* as an example, this complete color array is decomposed into several series of color slices. These cover all the colors of the encrypted target and serve as guides for a single-color device array. In contrast to the original *Iris* image, the product obtained from a

polarization direction of 90° is deliberately adopted for the target encryption. Based on the presented coupling theory model, each color slice is determined by the device grating period and ETL thickness, as revealed in Fig. 5b for blue, cyan, green, yellow, and red, respectively. Accordingly, with the introduction of a polarizer, the picture color changes significantly with the rotation of the optical axis, and the encrypted target can be reproduced only when it is rotated to 90°. This is completely distinguishable from the original image observed with the naked eye (Fig. 5c). The fore-polarization process, along with the multi-color polarization encryption facilitated by micro-LP-OLEDs, enables the double encryption of colorful images. To the best of our knowledge, this is the attempt at polarization-controlled chromo-encryption based on electroluminescent devices.

In summary, we demonstrate a controllable fabrication strategy for a dual-color emissive LP-OLED with an orthogonal polarization mode, by using a dielectric/metal nanograting-waveguide geometry and a combined cost-efficient method of laser interference lithography and vacuum thermal evaporation. The fine modulation of the ETL waveguide thickness combined with the subtle embedding of the Ag/MgF₂ nanograting geometry into the OLED devices coincidently satisfies the momentum matching of the grating structure and air space, thus selectively diffracting the desired sky-blue TE mode emission confined within the waveguide layer. Additionally, a green TM mode beyond the waveguide confinement frequency is emitted into the air-free space, forming a dual-color spatial pattern with orthogonal polarization. This is a tentative theoretical proposal to

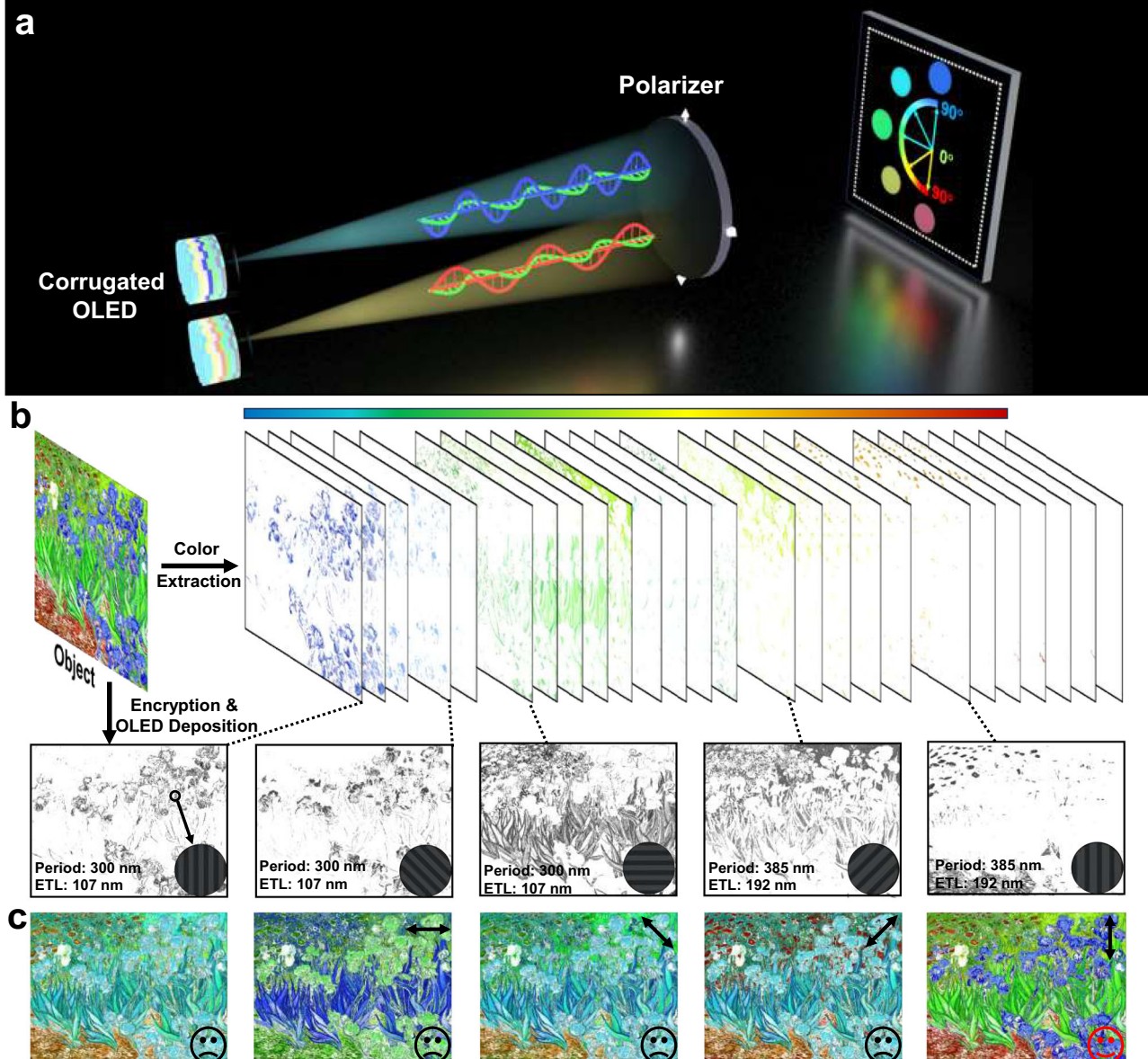

**Fig. 5 | Extended application of polarization and color-controlled chromo-encryption. a** Multi-color production achieved from dual-color emissive LP-OLEDs with orthogonal polarization mode. **b** Schematic of encryption process for colorful *Iris* picture that is decomposed into several series of color slices and achieved by several corrugated OLEDs with different periods and electron transport layer (ETL) thicknesses. **c** Different colorful *Iris* picture without and with polarizer in different directions. Sad expressions indicate a lack of matching success, while happy expressions indicate successful matching.

employ an electroluminescent linear polarization emission with orthogonal polarization for the dual encryption of colorful images. The diversity of degrees of freedom, such as the grating geometry, waveguide thickness, and material type, can enable a wealth of applications in solid-state colors, high-density optical data storage, and encryption.

## Methods

### Ag/MgF₂ nanograting fabrication on the glass substrate

Glass substrates were cleaned sequentially with deionized water, ethanol, and acetone. Prior to spin-coating the photoresist layers, the substrates were dried and exposed to ultraviolet-ozone for 20 min. After the glass was cooled to room temperature, the 145-nm-thick ultrathin layer of photoresist (AR-P3170) was spin-coated onto the glass substrate at 3000 r.p.m. for 60 s and then baked at 110 °C for 70 s

(Supplementary Fig. 10). A laser source with a wavelength of 343 nm was split into two pathways and eventually merged and interfered with the photoresist, thereby generating a grating pattern with a period of 300 nm after development (AR 300-47). It is worth noting that laser beam expansion and alignment are achieved by using two convex lenses, which determine the quality of the final grating. In addition, the size of the interference spot was adjusted according to the focal length. A laser beam with a dose of approximately -80 mJ cm$^{-2}$ was used, corresponding to a development time of 8 s. The template was cleaned using an $O_2$ plasma (120 W, 90 s) and loaded into a thermal evaporation chamber. MgF₂ grating was fabricated on cleaned template that was pre-deposited with 100-nm-thick MgF₂, which was formed using thermal evaporation at a pressure of $3 \times 10^{-4}$ Torr. Finally, a 25-nm-thick Ag layer was deposited on the MgF₂ nanograting via thermal evaporation to obtain the Ag/MgF₂ nanograting.

## Fabrication of OLEDs

The bottom-emitting OLEDs were fabricated on the $Ag/MgF_2$ grating substrate by thermal evaporation of the following materials in sequence: Al as anode; 1,4,5,8,9,11-hexaazatriphenylene-hexanitrile (HATCN) as hole injection layer; 1,1′-bis((di-4-tolylamino) phenyl) cyclohexane (TAPC) as hole transporting layer; 4,4′4″-tris(N-carbazolyl) triphenyl-amine (TCTA) as electron blocking layer; iridium (III)bis-(4,6-difluorophenyl-pyridinato-N,C²)-picolinate (FIrpic) doped N, N′-dicarbazolyl-3,5- benzene (mCP) or TCTA as emitting layer; 1,3,5-tri(m-pyridin-3-ylphenl) benzene (TmPyPb) as electron transporting layer; and LiF/Ag as cathode. The device structure was as follows: Ag (25 nm)/ LiF (0.8 nm)/TmPyPb (107 nm)/mCP:FIrpic (8%, 15 nm)/TCTA:FIrpic (8%, 15 nm)/TCTA (5 nm)/TAPC (40 nm)/ HAT-CN (10 nm)/Al (120 nm). The evaporation rate of the organic materials was $0.1$-$1 \, Å \, s^{-1}$, and that of metallic materials was $0.1$-$2 \, Å \, s^{-1}$, during the evaporation process. All the materials were evaporated at a pressure of $3 \times 10^{-4}$ pa。

## Characterization of OLEDs

The EQE–luminance–voltage curves and EL spectra for the planar and granting OLEDs were measured with a programmable source meter (Keithley 2400) and a spectrophotometer (Spectrascan PR650), which measures power per steradian per unit wavelength per unit area (in units of $W \, nm^{-1} \cdot sr^{-1} \cdot m^{-2}$). Due to the distinct directional emission of the corrugated OLED, breaking the traditional Lambertian emission mode, an integrating sphere was utilized to characterize the device EQE, aiming to collect all the photons emitted from the front, back, and sides of the device. The angular distribution of the EL was measured using a source meter (Keithley 2400), linear polarizer (GCL-0510), goniometer, and fiber-optic spectrometer (Ocean Optics S2000). SEM measurements were performed using a JEOL JSM-7500F scanning electron microscope. Atomic force microscopy (AFM) was performed using a Dimension Icon AFM (Bruker Corporation). The film thicknesses were measured using an XP-2 stylus profilometer.

## Theoretical Modeling

The optical reflectivity of the electrode, near-field intensity distribution, far-field modal dispersion, and OLED optical mode distribution were calculated using the FDTD method together with in-house generated codes, where the structural parameters of the grating were set to 300 nm according to the AFM measurements. In the simulation, the wavelength-dependent refractive index (n, k) of Ag was calculated using the Drude model, and the refractive index (n, k) of the other materials were measured using an ellipsometer within the wavelength region of 400–850 nm, from which material models were automatically generated and adopted in the FDTD. Owing to the periodicity of the nanograting, periodic boundary conditions in the X-direction and perfectly matched layer (PML) boundary conditions in the Y-direction were employed to optimize the grating structure. These boundary conditions were used to simulate the electric field distribution inside the device. To analyze the optical mode distribution within the device, the X-direction boundary condition should be set as PML, and the X-span of the device should be set to 20 μm to minimize the impact of dimensions on the simulation. The emission profile of the far-field Air mode is examined using the change far field index module in the far-field projections, with the refractive index of the far-field environment set to 1. The light source was configured as a dipole, and a transmission box module in the optical power section was used to monitor the emission power. The average power was calculated by collecting the power in the three dipole polarization directions to reduce computational errors. Changes in parameters such as the film thickness and grating period were implemented using the scanning feature with a scan accuracy of 1 nm, and the corresponding calculation results were obtained using a power monitor. The mesh accuracy was set as $\lambda/dx = 18$ in the overall architecture and a smaller mesh (dx,

dy = 1 nm) was added in regions of the OLED and set FDTD dimension to 2D. For modal dispersion in reciprocal space, MATLAB software was used (Supplementary Software 1). More details about the code used to build the model, we have added it to the accompanying Supplementary Data file.

## Reporting summary

Further information on research design is available in the Nature Portfolio Reporting Summary linked to this article.

## Data availability

The authors declare that the main data supporting the findings of this study are available within the article and its Supplementary Information files. The source data underlying Figs. 1b, 2a, b, g, 3c, e, f, g, 4f, Supplementary Fig. 2b, 3a, 5a, b, 6–8, 10 are provided in the Source Data files, and extra data are available from the corresponding author on request. Source data are provided with this paper.

## Code availability

The code for simulation of planar and corrugated OLED, are provided in the Supplementary Data files.

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

## Acknowledgements

N. L. would like to acknowledge the financial support from the National Natural Science Foundation of China (22275005). T. Z. would like to

acknowledge the financial support from the National Natural Science Foundation of China (62375007). The authors gratefully acknowledge the following persons: Prof. Quanfu An, and Prof. Yongzhe Zhang of Beijing University of Technology for providing the AFM and film thickness measurements, respectively.

## Author contributions

R.C. Experimental investigations, Device fabrication and characterization, Optical simulation, Experimental images, Writing-original draft. N.L. Conceptualization, Investigation, Methodology, Supervision, Writing-review & editing. T.Z. Conceptualization, Investigation, Methodology. All authors discussed the experiments and results.

## Competing interests

The authors declare no competing interests.
