## [Peer Review File · Nature Communications]

REVIEWER COMMENTS

Reviewer #1 (Remarks to the Author):

This paper reports the demonstration of a very efficient dual-colour OLED that has a high polarisation extinction ratio of 15.8 dB. The study includes a detailed theoretical analysis of the electromagnetic modes and emission from the phosphorescent microcavity OLED. This analysis is used to identify a suitable design that can achieve a high polarisation ratio of light emission.

The study of the structure is very detailed and carefully undertaken, and successfully identifies a suitable design to optimise the polarisation ratio of the emitted light. The device was then fabricated using interference lithography and thermal evaporation, and the performance of the resulting OLED was found to match well with calculations. The authors finally go on to present two novelties of the work – firstly that they can control the emission colour from the OLED by changing a polarising analyser in front of the device; and secondly, they propose using a display made from such OLEDs for colour-controlled chromo-encryption. This is an interesting idea, although they could perhaps comment on where this might see application.

Overall, I think this is an excellent scientific contribution to the field, and is suitable for publication in Nature Communications, after satisfactorily addressing the following comments.

1. There are a number of places in the manuscript where some further proofreading of the English is necessary to elevate it to the level needed for publication.
2. While the study looks to have been very carefully undertaken, I believe that more details are required in the methods section to explain clearly the experimental methods used. Currently the details included are insufficient for another group to be able to replicate the work. The authors should please add further detail in the explanation of the interference lithography process, including the thickness of the photoresist layer that they used, the fluence and exposure duration of the light exposure, and the developing time and parameters. Did the authors condition the laser beam to avoid spurious unintended interference effects?
3. It is also unclear what was the purpose of the oxygen plasma treatment - was this to remove a residual layer of photoresist in the structure?
4. I note that the electromagnetic simulation assumes a particular shape of grating (e.g. as indicated by the dotted lines in figure 2(i) and 2(j)). The authors should please comment on whether the grating shape was derived from experimental measurement. Did they consider adjusting the unit cell profile to control the light emission?
5. The authors should also please comment further on how the EQE was measured for the non-Lambertian emission patterns. How did they take account of the angular dependence of the emission?
6. I feel that the methods section on theoretical modelling is also lacking detail. The authors should provide more details on the FDTD simulation. It is best practice to publish the in-house generated codes used, particularly as this is a very important part of this publication.

Reviewer #2 (Remarks to the Author):

This paper by Chen and coworkers reports an interesting OLED device with a dual-color emission and an orthogonal polarization mode, theoretically providing its potential application in color-polarization encryption for colorful images. The optimized corrugated OLED devices in this work are constructed via low-cost laser interference lithography and vacuum thermal evaporation methods, to embed a D/M nanograting into OLED device. About this, the authors provide a detailed and reliable design principles, along with extensive simulation and analysis, aiming to explain the maximum extent localization into the organic waveguide layer and the following diffraction as much as possible into the air of TE-polarization mode, respectively.

Given the fact that linearly polarized light-emitting diodes have become an appealing functional expansion in polarization optics and optoelectronic applications and attract increasing attention, the construction of high performance LP-OLED with dual-color and orthogonal polarization emission shown in this paper is expected to have an impact on related fields, like naked-eye 3D displays, visible light communication and information anticounterfeiting and encryption and so on. Basically, experiments and theoretical considerations have been made rationally. I highly recommend this paper to be published in this journal, after revising the following comments.

- (1) Inverted device architecture was employed to construct the corrugated OLED devices. Could the author explain this?
- (2) Why was TE polarization mode selected as the target emission mode? Why not TM? The author should give a relevant introduction at the beginning or suitable position of this article.
- (3) The resulting corrugated OLED presented a relatively higher storage stability than that of the planar OLED. The authors gave a rough explanation, "This may be mainly ascribed to the hybrid microcavity effect on the exciton lifetime and the more uniform photons distribution". Detailed verification or explanation should be provided.
- (4) Main text, this demonstrates the green light was suppressed significantly, rather than was completely suppressed or disappeared, as displayed in Figure 3c. Otherwise, it contradicts the subsequent dual-color emission with orthogonal polarization characteristics. The author should give a more objective description about Figure 3C.
- (5) Proper proofreading is required for this manuscript, owing to the appearing English language, typos, and grammar errors, like "Since for those planar OLEDs, some photons are confined with the substrate and waveguide layer, as well as propagating along the organic/metal interface.²¹ As a result, the leak into air acts as background light, leading to complex emission profiles with scarcely polarization extinction ratio and low light extraction efficiency (LEE).²²" in abstract, "Bragg gratings efficiently promoted out-coupling by providing the TE waveguide light with additional momentum to couple to air" in title 3 of Results section, "The fore-polarization process, along with multicolor polarization encryption from micro-LP-OLED, facilitate a double encryption for colorful images." in title 6 of Results section.

Reviewer #3 (Remarks to the Author):

The work by Chen, Liang, and Zhai introduces an OLED with dominant emission of linearly polarized light based on a corrugated structure. While an OLED with corrugated structures is

not necessarily new, the efforts to make use of its polarization characteristics with dual color emission may be regarded novel.

The presentation of the work is well organized, with optical analysis, fabrication, experimental verification, and demonstration of examples. In this regard, Reviewer is in favor of publishing this manuscript after revision addressing the following:

(1) Some figures (and legends inside) are too small to be legible. Please revise them for better visibility and clear readability.

(2) In the experimental results, the efficiency of FIrPic-based OLEDs appears rather too low for all the cases studied in this work. Please provide a plausible explanation in the main text. (Optional) If possible, consider re-fabricating the device and try to improve the baseline performance.

(3) Emission characteristics of the corrugated OLED has specific angular profiles that may break the azimuthal symmetry often assumed in a regular, planar OLED. This may make a typical goniometric measurement, in which only a polar angle is varied, invalid. Please clearly indicate how authors taken this issue into account in the EL measurement.

(4) OLEDs built on a corrugated surface may suffer from relatively large leakage current. It is thus important to show J-V characteristics in a logarithmic scale.

(5) Please use "log", not "lg".

(6) Mostly well written, but there are grammatical errors or expressions that sound a bit unnatural. Please have the manuscript thoroughly proofread by English professional.

The Editor's comments (in black and italics), our response (in blue) and corresponding correction (in red)

Reviewer 1:

This paper reports the demonstration of a very efficient dual-colour OLED that has a high polarisation extinction ratio of 15.8 dB. The study includes a detailed theoretical analysis of the electromagnetic modes and emission from the phosphorescent microcavity OLED. This analysis is used to identify a suitable design that can achieve a high polarisation ratio of light emission.

The study of the structure is very detailed and carefully undertaken, and successfully identifies a suitable design to optimise the polarisation ratio of the emitted light. The device was then fabricated using interference lithography and thermal evaporation, and the performance of the resulting OLED was found to match well with calculations. The authors finally go onto present two novelties of the work – firstly that they can control the emission colour from the OLED by changing a polarising analyser in front of the device; and secondly, they propose using a display made from such OLEDs for colour-controlled chromo-encryption. This is an interesting idea, although they could perhaps comment on where this might see application.

Overall, I think this is an excellent scientific contribution to the field, and is suitable for publication in Nature Communications, after satisfactorily addressing the following comments.

(1) There are a number of places in the manuscript where some further proofreading of the English is necessary to elevate it to the level needed for publication.

Our response: Thank you for your kind remind. We have further proofread the English writing of the manuscript, and the corresponding revisions in grammar, spelling, and expression have been marked in Red.

(2) While the study looks to have been very carefully undertaken, I believe that more details are required in the methods section to explain clearly the experimental methods used. Currently the details included are insufficient for another group to be able to replicate the work. The authors should please add further detail in the explanation of the interference lithography process, including the thickness of the photoresist layer that they used, the fluence and exposure duration of the light exposure, and the developing time and parameters. Did the authors condition the laser beam to avoid spurious unintended interference effects?

Our response: Thank you for your reminder. Further detail in the explanation of the interference lithography process, are added in the Method Section, like “After the glass was cooled to room temperature, the 145-nm-thick ultrathin layer of photoresist (AP-R3170) was spin-coated onto the glass substrate at 3,000 r.p.m. for 60 s and then baked at 110 °C for 70 s (Supplementary Fig. S9).” and Supporting Figure 9 and corresponding description “The 145-nm-thick ultrathin positive photoresist films were spin-coated onto the cleaned substrate evenly (3000 r.p.m.) and the thickness can be tuned by rotation speed (Supplementary Fig. S9).” in Supporting Information for the thickness of the photoresist layer for the thickness of the photoresist layer; and “It is worth noting that laser beam expansion and alignment are achieved by using two convex lenses, which determine the quality of the final grating. In addition, the size of the interference spot was adjusted according to the focal

length. A laser beam with a dose of approximately $\sim 80 \text{ mJ/cm}^2$ was used, corresponding to a development time of 8 s." for the fluence and exposure duration of the light exposure and the developing time and parameters were also added into the Method Section. And the laser beam has not being conditioned.

Supplementary Figure 9: The summary of relationship between rotation speed and photoresist thickness.

(3) It is also unclear what was the purpose of the oxygen plasma treatment - was this to remove a residual layer of photoresist in the structure?

Our response: Thank you for this thoughtful comment. The use of oxygen plasma to treat the substrate is to remove residual photoresist in the groove, and therefore allowing MgF_2 layer to directly contact with glass substrate, as proved in **Figure 3b**. The main purpose is ascribed to enhance the following stability of the resulting D/M grating, and to maintain a relatively lower effective refractive index for a stronger light emission, as described in **Supplementary Figure 1b** in the calculation of far-field emission.

(4) I note that the electromagnetic simulation assumes a particular shape of grating (e.g. as indicated by the dotted lines in figure 2(i) and 2(j)). The authors should please comment on whether the grating shape was derived from experimental measurement. Did they consider adjusting the unit cell profile to control the light mission?

Our response: Thank you for your kind remind. The shape of the grating used in electromagnetic simulation in Figure 2i and 2j, is indeed resulted from the AFM measurement results of the D/M grating in the experiment (**Figure 1c**). This is exactly corresponding to the experimental results of the resulting OLED, as characterized in Figure 3b. According to your kind remind, detailed details about this section were added into the main text, as "The shape of the grating unit was adopted according to the AFM and cross-section SEM results for the D/M grating and OLED device, respectively.". Therefore, the shape of the grating unit in this current work was not considered and related research will be conducted in our near future.

(5) *The authors should also please comment further on how the EQE was measured for the non-Lambertian emission patterns. How did they take account of the angular dependence of the emission?*

Our response: We sincerely appreciate the referee for this valuable comment. Herein, we used an integrating sphere to measure the EQE values, in order to detect all the photons emitted from the front and back, as well as from sides of the device, thereby avoiding the conversion errors caused by the use of fiber optic detectors. According to your thoughtful remind, corresponding description was added into the main text, as “Due to the distinct directional emission of the corrugated OLED, breaking the traditional Lambertian emission mode, an integrating sphere was deliberately utilized to characterize the device J-V curves, EL spectra, and luminance-voltage-EQE curves, aiming to collect all the photons emitted from the front and back, as well as from sides of the device.⁵³⁻⁵⁴” Herein, corresponding references “[53] Isao, T., Shizuo, T. Precise Measurement of External Quantum Efficiency of Organic Light-Emitting Devices. *Jpn. J. Appl. Phys.* **43**, 7733 (2004). [54] Archer, E. et al. Accurate Efficiency Measurements of Organic Light-Emitting Diodes via Angle-Resolved Spectroscopy. *Adv. Opt. Mater.* **9**, 2000838 (2021).” were added into Reference Section. Furthermore, the angular dependence of the emission was measured by employing a high-precision spectrometer is connected to a fiber probe to capture the emission spectrum from the OLED at specific angles, as illustrated in **Supplementary Figure S8**. Detailed description like “Accordingly, for the optimized LP-OLEDs, the TE polarized mode presented a sky-blue color, with a θ of 90° , as shown in **Fig. 4a** and **Supporting Video**; whereas, the intensity of sky-blue TE polarized light dropped dramatically as the polarizer was rotated to 0° , ultimately displaying a green TM polarized light (**Fig. 4b**).” in original version. Therefore, these two parameters were characterized through different measurement methods. Detailed explanation and measurement about angular dependence of the emission were also added into Supporting Information, like “Angle-resolved emission spectroscopy is used to characterize Air mode dispersion under TE and TM polarization. On the one hand, the one-dimensional grating introduces grating momentum only in the horizontal direction, corresponding to the measurement of the polar angle. On the other hand, the polarization emission in the normal direction (with polar angle and azimuth angle equal to 0°) was obtained through optimization strategies. As a result, by measuring only the polar angle, we can easily obtain an effective mode dispersion spectrum. The TE light is filtered through the polarizer and the TM light is filtered out by making the fast axis of the polarizer parallel to the grating line of the 1-D grating structure. Similarly, the measurement of TM light can be completed by rotating 90° at this position. The probe is connected to a high-precision spectrometer and can be used to detect spectra at specific angles.”

(6) *I feel that the methods section on theoretical modelling is also lacking detail. The authors should provide more details on the FDTD simulation. It is best practice to publish the in-house generated codes used, particularly as this is a very important part of this publication.*

Our response: Thank you for you remind. More details about the FDTD simulation were added into the main text and Method Section, like “These boundary conditions were used to simulate the electric field distribution inside the device. To analyze the optical mode distribution within the device, the X-direction boundary condition should be set as PML, and the X-span of the device should be set to $20\ \mu\text{m}$ to minimize the impact of dimensions on the simulation. The emission profile of the far-field Air mode is examined using the "change far field index" module in the far-field projections, with the refractive index of the far-field environment set to 1. The light source was configured as a dipole, and a transmission box module in the optical power section was used to monitor the emission power. The

average power was calculated by collecting the power in the three dipole polarization directions to reduce computational errors. Changes in parameters such as the film thickness and grating period were implemented using the scanning feature with a scan accuracy of 1 nm, and the corresponding calculation results were obtained using a power monitor. The mesh accuracy was set as $\lambda/dx = 18$ in the overall architecture and a smaller mesh ($dx, dy = 1$ nm) was added in regions of the OLED and set FDTD dimension to 2D.”. More details about the code used to build the model, we have added it to the accompanying Supplementary Data file. The relevant parameters used in the code can be found in the main text.

Reviewer #2:

This paper by Chen and coworkers reports an interesting OLED device with a dual-color emission and an orthogonal polarization mode, theoretically providing its potential application in color-polarization encryption for colorful images. The optimized corrugated OLED devices in this work are constructed via low-cost laser interference lithography and vacuum thermal evaporation methods, to embed a D/M nanograting into OLED device. About this, the authors provide a detailed and reliable design principles, along with extensive simulation and analysis, aiming to explain the maximum extent localization into the organic waveguide layer and the following diffraction as much as possible into the air of TE-polarization mode, respectively.

Given the fact that linearly polarized light-emitting diodes have become an appealing functional expansion in polarization optics and optoelectronic applications and attract increasing attention, the construction of high-performance LP-OLED with dual-color and orthogonal polarization emission shown in this paper is expected to have an impact on relate fields, like naked-eye 3D displays, visible light communication and information anticounterfeiting and encryption and so on. Basically, experiments and theoretical considerations have been made rationally. I highly recommend this paper to be published in this journal, after revising the following comments.

(1) Inverted device architecture was employed to construct the corrugated OLED devices. Could the author explain this?

Our response: We thank the reviewer for this thoughtful comment. Conventional corrugated architecture was not employed, because of its rather low device performance. Even at the case of the identical operating condition controlled with that of inverted corrugated device (seeing the following Figure 1 and Table 1). Therefore, considering the poor device performance, inverted architecture was adopted, and relatively higher performance with EQE of 2.25% was realized, fortunately. Maybe, detailed research work about the big performance difference between these two types of devices, should be focused on our near future work.

Figure 1: Luminance-EQE-Voltage curves for normal-structure corrugated OLED with 107-nm-thick ETL.

Table 1. Summary performance parameters of conventional corrugated OLED with ETL thickness of 107 nm.

Device	$V_{\text{turn-on}}$ (V)	$\eta_{\text{C,max}}$ (cd/A)	$\eta_{\text{p,max}}$ (lm/W)	EQE MAX (%)
normal-structure corrugated OLED	13	0.16	0.026	0.07

(2) Why was TE polarization mode selected as the target emission mode? Why not TM? The author should give a relevant introduction at the beginning or suitable position of this article.

Our response: Thank you very much for your kind remind. On one hand, the SPP mode introduces higher losses for TM polarization when compared to the TE waveguide mode. On the other hand, achieving energy localization is relatively simpler with the TE waveguide mode by adjusting the thickness of the ETL. According to your thoughtful comment, detailed introduction was added into the Introduction section, as “Moreover, for a planar device, the TM waveguide mode apparently excites the surface plasmon polariton (SPP) mode that is distributed near the metal surfaces and decays quickly toward the middle of the device, causing a large emission loss. By contrast, the TE waveguide mode polarized parallel to the thin-film surface with different directional propagations can be extracted significantly by constructing a corrugated structure.”.

(3) The resulting corrugated OLED presented a relatively higher storage stability than that of the planar OLED. The authors gave a rough explanation, “This may be mainly ascribed to the hybrid microcavity effect on the exciton lifetime and the more uniform photons distribution”. Detailed verification or explanation should be provided.

Our response: Thank you for your kind remind. Base on literature (Qi, P. et al. *Light Sci. Appl.* **11**, 176 (2022); Laitz, M. et al. *Nat. Commun.* **14**, 2426 (2023)), we realized that the introduction of an optical microcavity serves to enhance the interaction between light and matter, and therefore, significantly diminishing the exciton lifetime, mitigating the accumulation of excitons, and weakening non-radiative exciton transitions. Additionally, the exciton lifetime plays a critical role in determining the overall device lifetime, because the prolonged lifetimes of electron-injected triplet excitons could give rise to various non-radiative transitions, including triplet-triplet, triplet-singlet, and triplet-polaron annihilation processes. These non-radiative transitions release energy in the form of heat or other forms, thereby impacting the device's lifetime. Consequently, the corrugated OLED with enhanced light-matter interaction offers enhanced stability compared to planar OLEDs. Herein, reliable references “[55] Qi, P. et al. Giant excitonic upconverted emission from two-dimensional semiconductor in doubly resonant plasmonic nanocavity. *Light Sci. Appl.* **11**, 176 (2022); [56] Laitz, M. et al. Uncovering temperature-dependent exciton-polariton relaxation mechanisms in hybrid organic-inorganic perovskites. *Nat. Commun.* **14**, 2426 (2023).” were added into corresponding position, and comprehensive characterization, along with detailed explanation will be conducted in our near future work.

(4) *Main text, this demonstrates the green light was suppressed significantly, rather than was completely suppressed or disappeared, as displayed in Figure 3c. Otherwise, it contradicts the subsequent dual-color emission with orthogonal polarization characteristics. The author should give a more objective description about Figure 3C.*

Our response: Thank you for your kind remind. In order to eliminate ambiguity, we revised the statement from “Contrast to the reference OLED with 60-nm-ETL, both planar and corrugated OLED with 107-nm-ETL, presented a significantly narrowed emission spectra with a distinguishable EL peak at $\lambda \sim 470$ nm and almost a disappeared peak at $\lambda \sim 500$ nm” to “ Contrast to the reference OLED with 60-nm-ETL, both planar and corrugated OLED with 107-nm-ETL, presented a significantly narrowed emission spectra with a distinguishable EL peak at $\lambda \sim 470$ nm and a significantly suppressed peak at $\lambda \sim 500$ nm.”.

(5) *Proper proofreading is required for this manuscript, owing to the appearing English language, typos, and grammar errors, like “Since for those planar OLEDs, some photons are confined with the substrate and waveguide layer, as well as propagating along the organic/metal interface.²¹ As a result, the leak into air acts as background light, leading to complex emission profiles with scarcely polarization extinction ratio and low light extraction efficiency (LEE).²² ” in abstract, “Bragg gratings efficiently promoted out-coupling by providing the TE waveguide light with additional momentum to couple to air ” in title 3 of Results section, “The fore-polarization process, along with multicolor polarization encryption from micro-LP-OLED, facilitate a double encryption for colorful images. ” in title 6 of Results section.*

Our response: We thank the referee for this comment. We have revised the language and grammar issues in the revised manuscript. The first sentence was altered into “ In these planar OLEDs, some

photons are confined with the substrate and waveguide layer, while propagating along the organic/metal interface.²¹ As a result, the leakage into the air contributes to the background light, leading to complex emission profiles with a scarcely noticeable polarization extinction ratio and low light extraction efficiency (LEE).²² This ratio is defined as $ER_{TM/TE}=10\log I_{TM}/I_{TE}$, where I_{TM} and I_{TE} are intensities of transverse-magnetic (TM) and transverse-electric (TE) polarized emission at the same wavelength. ”. The second was modified into “The Bragg gratings efficiently promoted out-coupling by providing the TE waveguide light with additional momentum to couple into the air space”. The third sentence was altered to “The fore-polarization process, along with the multicolor polarization encryption facilitated by micro-LP-OLED, enables the double encryption of colorful images.”

Reviewer #3:

The work by Chen, Liang, and Zhai introduces an OLED with dominant emission of linearly polarized light based on a corrugated structure. While an OLED with corrugated structures is not necessarily new, the efforts to make use of its polarization characteristics with dual color emission may be regarded novel.

The presentation of the work is well organized, with optical analysis, fabrication, experimental verification, and demonstration of examples. In this regard, Reviewer is in favor of publishing this manuscript after revision addressing the following:

(1) Some figures (and legends inside) are too small to be legible. Please revise them for better visibility and clear readability.

Our response: We thank the reviewer for this kind remind. Necessary modifications to several of the images, like AFM in Figure 1, Figure 2, Figure 3a 3g 3i, Figure 4a 4b 4c 4d 4f and Figure S2a 2b, were revised to enhance their quality and clarity.

(2) In the experimental results, the efficiency of FIrPic-based OLEDs appears rather too low for all the cases studied in this work. Please provide a plausible explanation in the main text. (Optional) If possible, consider re-fabricating the device and try to improve the baseline performance.

Our response: Thank you very much for your kind remind. An EQE of 15.7% was achieved in the conventional planar devices with a reference architecture of ITO/HAT-CN (10nm)/TAPC (40nm)/TCTA (5nm)/TCTA:FIrpic (8%, 15nm)/mCP:FIrpic (8%, 15nm)/TmPyPb (40 nm)/LiF (0.8nm)/Al (120nm) (**Supplementary Figure S6a**). These parameters reach the typical performance level of FIrPic-based OLEDs (Kim, J. et al. Sci. Rep. 4, 7009 (2014); Y, C. et al. Appl. Phys. Lett. 100, 213301 (2012)). Herein, for the planar OLEDs in this work, 25-nm-Ag and 107-nm-TmPyPb were adopted as the bottom cathode and ETL, respectively. Therefore, a metal reflection waveguide, along with additional SPP and sever TE waveguide modes were established, thus significantly reducing the photonic emission. Therefore, to reducing ambiguity, the current density-voltage-EQE and current-voltage-luminance curves for the conventional planar devices (ITO/HAT-CN

(10nm)/TAPC (40nm)/TCTA (5nm)/TCTA:FIrpic (8%, 15nm)/mCP:FIrpic (8%, 15nm)/TmPyPb (40 nm)/LiF (0.8nm)/Al (120nm)) were added into the Supporting Information (Supplementary Figure 6). Relevant description, like “Notably, the optimized corrugated OLEDs exhibited a lower EQE than conventional planar ITO-based OLED (Supplementary Fig. S6), which resulted from the employed 25-nm-Ag cathode and ultrathick 107-nm-ETL; whereas, an outstanding TE polarization mode emission was realized. Detailed characterization and analyses are presented in the following sections.” was added into the main text.

Supplementary Figure 6: (a) Current density -EQE- Voltage curves, and (b) Current- Luminance- Voltage curves for ITO-based OLEDs. The device structure is ITO/ HAT-CN (10nm)/TAPC (40nm) /TCTA (5nm)/TCTA:FIrpic (8%, 15nm)/mCP:FIrpic (8%, 15nm)/ TmPyPb (40 nm)/LiF (0.8nm) /Al (120nm).

(3) Emission characteristics of the corrugated OLED has specific angular profiles that may break the azimuthal symmetry often assumed in a regular, planar OLED. This may make a typical goniometric measurement, in which only a polar angle is varied, invalid. Please clearly indicate how authors taken this issue into account in the EL measurement.

Our response: Thank you for your kind remind and we apologize that the measurement methods for EL, and EQE spectra, as well as for the polarization performance, were not described clearly. Similar question has also been proposed by Reviewer 1. Herein, we used an integrating sphere to measure the EQE values, to detect all the photons emitted from the front and back, as well as from sides of the device, thereby avoiding the conversion errors caused using fiber optic detectors. According to your thoughtful remind, corresponding description was added into the Method Section, as “Due to the distinct directional emission of the corrugated OLED, breaking the traditional Lambertian emission mode, an integrating sphere was deliberately utilized to characterize the device J-V curves, EL spectra, and luminance-voltage-EQE curves, aiming to collect all the photons emitted from the front and back, as well as from sides of the device.⁵³⁻⁵⁴” Furthermore, the angular dependence of the emission was measured by employing a high-precision spectrometer is connected to a fiber probe to capture the emission spectrum from the OLED at specific angles, as illustrated in Supplementary Figure S8. Detailed description like “Accordingly, for the optimized LP-OLEDs, the TE polarized mode presented a sky-blue color, with a θ of 90° , as shown in Fig. 4a and Supporting Video; whereas, the intensity of sky-blue TE polarized light dropped dramatically as the polarizer was rotated to 0° , ultimately displaying a green TM polarized light (Fig. 4b).” in original version. Additionally, numerical analysis proves that the emission profile has horizontal and vertical symmetry and indicates

that mode dispersion is more effectively studied using the polar angle instead of the azimuth angle. Further detail in the explanation was added in the main text, like “The emission profile of the corrugated OLED displayed both horizontal and vertical symmetry, resulting from the symmetry of grating structure and exciton recombination randomness.⁵⁷⁻⁵⁸” Herein, reliable references “[53] Isao, T., Shizuo, T. Precise Measurement of External Quantum Efficiency of Organic Light-Emitting Devices. *Jpn. J. Appl. Phys.* **43**, 7733 (2004). [54] Archer, E. et al. Accurate Efficiency Measurements of Organic Light-Emitting Diodes via Angle-Resolved Spectroscopy. *Adv. Opt. Mater.* **9**, 2000838 (2021). [57] Hu, C. et al. Source-configured symmetry-broken hyperbolic polaritons. *eLight*. **3**, 14 (2023). [58] Wang, M. et al. Spin-orbit-locked hyperbolic polariton vortices carrying reconfigurable topological charges. *eLight*. **2**, 12 (2022).” were added into corresponding position. In summary, these two parameters were characterized through different measurement methods.

(4) OLEDs built on a corrugated surface may suffer from relatively large leakage current. It is thus important to show J-V characteristics in a logarithmic scale.

Our response: Thank you for your thoughtful suggestion. J-V characteristics in a logarithmic scale for corrugated and planar OLED with 107-nm-ETL were added in Supporting Figure S5. Relevant description has been altered in the main text, as “The EL spectra, luminance-voltage-EQE curves, angle-resolved EL intensity in the normal direction at an emission peak of 470 nm wavelength, corresponding TE-and TM-polarized mode emission, and J-V curves were measured using the same setup, as shown in Fig. 3c-3l and Supporting Figure S5. Interestingly, the corrugated OLED exhibited a significantly enhanced current density than that of planar OLED with 107-nm-ETL.”.

Supplementary Figure 5: Current density-Voltage curves for corrugated and planar OLED.

(5) Please use "log", not "lg".

Our response: We thank the referee for this kind remind. “lg” has been altered into “log” in the full main text.

(6) Mostly well written, but there are grammatical errors or expressions that sound a bit unnatural. Please have the manuscript thoroughly proofread by English professional.

Our response: Thank you for your kind remind. We have further proofread the English writing of the manuscript, and the corresponding revisions in grammar, spelling, and expression have been marked in **Red**.

REVIEWERS' COMMENTS

Reviewer #1 (Remarks to the Author):

I am content that the authors have addressed all of my previous comments satisfactorily. I rate the revised manuscript and supporting information as suitable for publication in Nature Communications.

Reviewer #2 (Remarks to the Author):

The comments/concerns have been well addressed and this work will inspire readers and researchers in the related research community.

Reviewer #3 (Remarks to the Author):

Authors' revision has made several issues raised in the initial review clarified. Reviewer believes the manuscript is now almost ready for publication; however, a minor revision addressing the following is recommended before final acceptance.

(1) In Figure 1a, indicate the layer thickness at least for those with the fixed values (e.g. Ag, Al, and MgF₂). Change the white color of the words like "organic", "Ag", and "MgF₂" with a different color leading to a better contrast.

(2) In Figure 1b, consider including device layer structures under study. In a similar context, its caption "Molecular structure and EL spectrum of conventional OLED" is not clear at all. Along with addition of the device structures studied in the present work, clearly indicate from which the presented spectrum was obtained. Furthermore, add "wavelength (nm)" in the horizontal axis of the spectrum graph.

Throughout the manuscript, authors should be more specific when mentioning "conventional OLED." For example, "conventional OLEDs with the same Irpic-based EML in a bottom-emission, inverted geometry" will be much clearer to readers than "conventional OLEDs".

(3) In Figure 2g, the vertical axis indicates the mode fraction in percentage, but it seems it is presented in 0 to 1 scale, instead of 0 to 100%.

(4) The caption of Figure 2h should provide more specific information on what is exactly shown in the diagram.

(5) In Figure 3, it is not clear why authors chose 60-nm-thick ETL for a reference device. It would be helpful if authors add more explanation in the main text regarding why it was chosen and why it exhibited a different behavior compared to the other devices under study. It would be better if authors can provide explanation in connection to what is shown in Figure 2.

The Editor's comments (in black and italics), our response (in blue) to Reviewer #3 and corresponding correction (in red)

Reviewer #1:

I am content that the authors have addressed all of my previous comments satisfactorily. I rate the revised manuscript and supporting information as suitable for publication in Nature Communications.

Reviewer #2:

The comments/concerns have been well addressed and this work will inspire readers and researchers in the related research community.

Reviewer #3:

Authors' revision has made several issues raised in the initial review clarified. Reviewer believes the manuscript is now almost ready for publication; however, a minor revision addressing the following is recommended before final acceptance.

(1) In Figure 1a, indicate the layer thickness at least for those with the fixed values (e.g. Ag, Al, and MgF2). Change the white color of the words like "organic", "Ag", and "MgF2" with a different color leading to a better contrast.

Our response: Thank you for your kind remind. We have made necessary modifications to **Figure 1a** to improve the contrast.

(2) In Figure 1b, consider including device layer structures under study. In a similar context, its caption "Molecular structure and EL spectrum of conventional OLED" is not clear at all. Along with addition of the device structures studied in the present work, clearly indicate from which the presented spectrum was obtained. Furthermore, add "wavelength (nm)" in the horizontal axis of the spectrum graph. Throughout the manuscript, authors should be more specific when mentioning "conventional OLED." For example, "conventional OLEDs with the same FIrpic-based EML in a bottom-emission, inverted geometry" will be much clearer to readers than "conventional OLEDs".

Our response: Thank you for your kind remind. We indeed have not make a clear description about the conventional OLED, and appropriate modifications have been made, including the caption of Figure 2b to "Molecular structure and EL spectrum of conventional OLED with the same FIrpic-based EML; the insert presents the device architecture⁴⁵", as well as its corresponding description has been revised to "The chemical structures of emitting materials are shown in **Fig. 1b** and corresponding EL spectrum for conventional OLEDs employing the same FIrpic-based EML adopting indium tin oxide (ITO) as anode ranges from 440 to 650 nm, which exhibits a FWHM of 67 nm and a dual-peak at 470 and 500 nm, respectively." Additionally, the "conventional OLED" throughout this manuscript has been added into "conventional OLED with the same FIrpic-based EML based on ITO in **Figure 1b**"

(3) *In Figure 2g, the vertical axis indicates the mode fraction in percentage, but it seems it is presented in 0 to 1 scale, instead of 0 to 100%.*

Our response: Thank you for your kind remind. We indeed make a mistake and correct vertical axis presented in 0 to 100 has been added in **Figure 2g**.

(4) *The caption of Figure 2h should provide more specific information on what is exactly shown in the diagram.*

Our response: Thank you for your kind remind. According to your thoughtful comment, the caption of Figure 2h has been changed to "Schematic of planar OLED light emission with a gradual increase in emission angle as a more waveguide thickness, for a certain wavelength of 470 nm".

(5) *In Figure 3, it is not clear why authors chose 60-nm-thick ETL for a reference device. It would be helpful if authors add more explanation in the main text regarding why it was chosen and why it exhibited a different behavior compared to the other devices under study. It would be better if authors can provide explanation in connection to what is shown in Figure 2.*

Our response: We sincerely appreciate the referee for this valuable comment. According to **Figure 2g**, it clearly demonstrates that both a 60-nm-thick ETL and a 107-nm-thick ETL effectively suppress TE air-mode emission, with a higher localization of waveguide modes for OLED based on 107-nm-thick ETL to achieve a stronger TE light emission. Therefore, in order to further conform this

viewpoint, we selected a 60-nm ETL thickness as the reference OLED in our experiment. As a result, corresponding conclusion has been derived from **Figure 3c** illustrating that the 107-nm-thick ETL indeed achieves a stronger TE light emission. According to your kind remind, relevant description like “According to **Figure 2g**, it clearly demonstrated that both a 60-nm-thick ETL and a 107-nm-thick ETL effectively suppress TE air-mode emission, with a higher localization of waveguide modes for OLED based on 107-nm-thick ETL. Therefore, planar inverted OLEDs with an ETL thickness of 107 nm and 60 nm were also fabricated for comparison and were named as planar and reference OLED, respectively.” and “These results indicate that stronger TE waveguide localization, rather than air mode suppression, is more beneficial for the emission of TE polarized light.” have been added in corresponding position.